# Patient and retina specialists' preferences in neovascular age-related macular degeneration treatment. A discrete choice experiment

**Roberto Gallego-Pinazo[1], Begoña Pina-Marin[2], Marta Comellas[3], Susana Aceituno[3], Laia Gómez-Baldó[4], Carles Blanch[5]\*, on behalf of the AMD-Manage investigators[¶]**

**1** Unit of Macula, Oftalvist Clinic, Valencia, Spain, **2** Department of Ophthalmology, Hospital dos de Maig, Barcelona, Spain, **3** Outcomes'10, Castellón de la Plana, Spain, **4** Medical Department, Novartis Farmacéutica S. A., Barcelona, Spain, **5** Health Economics & Market Access. Novartis Farmacéutica S.A., Barcelona, Spain

¶ Membership of AMD-Manage investigators is listed in the Acknowledgments.
\* carles.blanch@novartis.com

**Data Availability Statement:** All relevant data are within the manuscript and its Supporting Information files.

## Abstract

### Introduction and objective

Neovascular age-related macular degeneration (nAMD) leads to severe and permanent visual impairment, significantly impacting patients' quality of life and functional independence. Although treatment with anti- vascular endothelial growth factor (VEGF) prevents and, in some cases, reverses visual damage, the need for frequent monitoring visits and intravitreal injections represents a significant burden on patients, caregivers and retina specialists.

### Objective

To elicit preferences for nAMD treatment characteristics from the perspectives of patients and retina specialists.

### Method

A discrete choice experiment was conducted. Participants (patients > 50 years with nAMD receiving anti-VEGF drugs for at least 2 years and without previous experience with anti-VEGF and retina specialists working in the Spanish National Healthcare System) were asked to select one of two hypothetical treatments resulting from the combination of five attributes (effects on visual function, effects on retinal fluid, treatment regimen, monitoring frequency, and cost); their levels were identified by reviewing the literature and two focus groups. The relative importance (RI) given to each attribute was estimated using a mixed logit model. The marginal rates of substitution (MRS) were calculated taking cost as the risk attribute.

**Funding:** The funder provided support in the form of salaries for authors [CB, LGB], but did not have any additional role in the study design, data collection and analysis, decision to publish, or preparation of the manuscript. The specific roles of these authors are articulated in the 'author contributions' section.

**Competing interests:** Two of the authors of the manuscript (LGB and CB) are Novartis employees, however this does not alter their adherence to PLOS ONE policies on sharing data and materials.

## Results

A total of 110 patients (P) [aged 79.0 (SD:7.4) years; 57.3% women; 2.3 (SD:0.7) years with nAMD; 2.1 years (SD:0.1) in treatment] and 66 retina specialists (RS) participated in the study. Participants gave greater RI to improvements in their visual function [60.0% (P); 52.7% (RS)], lower monitoring frequency [20.2% (P); 27.1% (RS)] and reduction in retinal fluid [9.8% (P); 13.0%(RS)]. Patients and retina specialists would agree to an increase in cost by 65.0% and 56.5%, respectively, in exchange for improvements of visual function; and 25.5% and 43.3% on delaying monitoring frequency by one month.

## Conclusions

Efficacy of treatment, in terms of visual function improvements, is the main driver for treatment election for both patients and retina specialists. Treatment monitoring requirements are also considered, mainly from the retina specialist's perspective. These results suggest that the use of more efficacious anti-VEGF agents with a longer duration of action may contribute to aligning treatment characteristics with patients/specialists' preferences. A better alignment would facilitate better disease management, fulfilling the unmet needs of patients and retina specialists.

## Introduction

Age-related macular degeneration (AMD) causes progressive loss of central vision [1]. Late AMD may result in severe and permanent visual impairment and legal blindness, significantly impacting patients' quality of life and functional independence [1]. The neovascular form of AMD (nAMD) accounts for 10% of AMD cases [2, 3]. In Europe the estimated prevalence of AMD will rise from 2.7 million people in 2016 to 3.9 million by 2040 [4].

During the last decade, management of nAMD has improved with the development of vascular endothelial growth factor (VEGF) inhibitors [5, 6]. These drugs reduce exudation from the leaky vessels and improve retinal morphology, leading to gains in visual acuity [7]. All are administered by intravitreal injection but differ according to monitoring and injection schemes.

Even when AMD does not lead to blindness, there might be a strong negative impact on independence and quality of life. A cross-sectional study revealed that AMD negatively impacted on day-to-day patients' activities and was associated with negative emotions such as fear, sadness, frustration, and depression [8]. Nearly one-third of patients with advanced nAMD reported a fall or accident in the previous two years due to their vision impairment and needed to be hospitalized as a result of a fall [8]. From the patient perspective, the treatment itself (having injections, frequency of injections and possible injection-related side effects), treatment cost and finding the right treatment options (information on choosing the best option) were the main barriers in managing the disease [9]. Anti-VEGF treatment has been shown to prevent and, in some cases, reverse visual decline caused by nAMD. However, the need for frequent monitoring visits and intravitreal injections lead a significant burden on patients, caregivers and retina specialists [10]. For this reason, new treatments, in addition to focusing on better disease control, aim to reduce the frequency of visits, injections and, therefore, the use of resources.

Given the advanced age of the affected population and the high impact of the disease on patients' lives and the burden of care placed on retina specialists, it is necessary to consider

several factors including patient profile, disease characteristics, drug access, healthcare resources available, management protocols, and healthcare burden, among others [11, 12]. The promotion of shared decision-making and incorporation of patient preferences in the disease management decision could improve the effectiveness of healthcare interventions by increasing patient satisfaction and improving adherence to treatments. Conjoint analysis methods have been applied successfully to measuring preferences for a diverse range of health applications [13], including ophthalmologic conditions such as glaucoma [14, 15], diabetic retinopathy [16] and AMD [16–19, 20]. Discrete choice experiments (DCE) in particular have become the most frequently used approach in health care [21]. DCE is a stated preference method based on two assumptions: 1) interventions or treatment can be described in terms of a conjoint set of attributes, and 2) the priority given to the intervention or treatment by an individual depends on the nature and level of the attributes, which means that individuals will always choose the alternative with the highest level of expected utility. A DCE presents a reasonably straightforward task and one which more closely resembles a real-world decision.

Although several studies have explored preferences from patients with AMD perspective, none of them have included the retina specialist perspective. Given the high burden of managing AMD for both patients and retina specialists, explore different stakeholders' preferences is crucial to understanding disease management. Therefore this study aims to elicit preferences for treatment characteristics in nAMD by including the perspectives of both patients and retina specialists.

## Methodology

### Study participants

Patients over the age of 50 years with nAMD receiving anti-VEGF drugs for at least 2 years and retina specialists working in the Spanish National Healthcare System (SNHS) were invited to participate in the study. They were selected from the universe of the AMD-MANAGE study (patients recruited from 20 public and private tertiary hospitals from different Spanish regions that met the following selection criteria: adult naïve (no previous exposure to anti-VEGF treatment) patients ≥50 years with confirmed nAMD diagnosis who started anti-VEGF treatment between November 1st, 2016 and February 28th, 2017, with a follow-up of 24 months and not participating in any other clinical study) (S1 Table) [22].

Recruitment of patients and retina specialists and data collection took place between November 2018 and January 2019 at 20 hospitals in the SNHS. All patients and retina specialists provided written informed consent to participate in the study.

The minimum sample size for analyzing main effects was estimated to be 42 nAMD patients and 42 retina specialists, based on Orme's rule-of-thumb $\frac{nta}{c} \geq 500$, where n = number of participants, t = number of choice tasks, a = number of alternatives per choice task, and c = maximum number of levels within an attribute [23].

### Discrete choice experiment

A discrete choice experiment (DCE) was conducted in accordance with International Society for Pharmacoeconomics and Outcomes Research (ISPOR) good practice recommendations for conjoint analysis in healthcare [13, 21]. Participants were asked to select one of two hypothetical treatments that resulted from combining a series of previously defined attributes (characteristics) and their levels (possible values of the attribute). Pairs of alternatives were then presented to the participants, who chose one of the two options each time.

The DCE results provide information on the relative importance (RI) of the different attributes and the rate at which respondents are willing to trade one attribute for preferred levels of another attribute (marginal rates of substitution, MRS) [24].

**Selection of attribute and levels.** A literature review was conducted to identify the potential attributes and levels to be included in the DCE. Key terms related to the disease ("age-related macular degeneration", "AMD", macular degeneration [MeSH]), treatment ("treatment", "medication) and stated-preferences studies ("conjoint", "conjoint analysis", "conjoint measurement", "conjoint studies", "discrete choice experiments", "DCE", "discrete choice modeling", "preference studies", Patient Preferences [MeSH]) joined by Booleans operators "or" and "and" were used to search in MedLine/PubMed, Cochrane Library, Institute for Scientific Information Web Of Knowledge (ISI WOK) and SCOPUS databases. Studies published until May 2018 that assessed patient or retina specialists' preferences for AMD treatment attributes and/or their willingness to pay for gaining health benefits or avoiding side effects were selected.

Two focus groups to define the set of attributes and levels to be included in the DCE were conducted, one with patients (n = 4 patients, 100% women, range age 45 to 70 years, 100% in anti-VEG treatment and one with experts in AMD management (n = 4 retina specialists and n = 2 hospital pharmacists working in tertiary hospitals). Patients with AMD who participated in the focus group were identified by the patient advocacy group (Mácula Retina); the study coordinator selected experts based on their expertise in AMD management. Participants in the focus groups discussed the validity and relevance of the potential attributes and levels identified in the literature review. Moreover, they completed the list with those attributes and levels not previously described in the literature but important from their perspective. Attributes were ranked from most to least important based on their preferences. The interpretation of the qualitative analysis and the analysis of the ranking exercises allowed to narrow down the list of attributes. Additionally, attributes and levels were tested to check for any problems in interpretation and face validity.

During the focus group with experts, consensus regarding the attributes/levels to be included in the DCE was achieved. As a result, five attributes were selected based on their relevance from patients' and experts' perspectives. Following ISPOR recommendation, levels of each attribute were limited to four, extreme values that may cause a grounding effect or the use of ranges to define attributes were avoided, and unplausible profiles were excluded [21] (Table 1).

**Construction of choice tasks and survey design.** A factorial design (orthogonal main-effect matrix) was applied to define the final scenarios, derived from combinations of the selected attribute levels. This method was used to ensure an orthogonal (each attribute level appears an equal number of times as all other attribute levels) and balanced (each level appears an equal number of times within an attribute) design [13, 21], yielding equally robust results for all levels. The combination of attributes and levels resulted in 18 hypothetical choice sets. These 18 choice sets were divided into two questionnaires containing nine choice sets each by using a blocked design [13]. Respondents were randomly assigned to a block and answered the choice questions in the block. Blocking promotes response efficiency by reducing the necessary cognitive effort for each respondent [13]. Additionally, we included a dominant choice set in the questionnaire to test for rationality (i.e., a choice set including one treatment profile characterized by logically preferable levels on all attributes). The dominant choice set was an extra choice set (i.e., 10 choice sets were created).

The patient questionnaire also collected their sociodemographic (age, sex) and clinical characteristics (time from nAMD diagnosis, time from anti-VEGF therapy and treatment regimen), and the choice sets included text and pictograms to facilitate understanding as most of the patients had low vision (Fig 1).

**Table 1. Attributes and levels included in the DCE.**

| Attribute | Level |
|---|---|
| **Effect on visual function** (best-corrected visual acuity improvements from baseline) | Stable (no changes) |
| | Improvement of more than 5 letters |
| | Improvement from 1–5 letters |
| **Effects on retinal fluid** (change in intraretinal fluid volume) | Reduction |
| | Resolution |
| | No changes |
| **Treatment regimen** (treatment plan included dosage, schedule, and duration of treatment) | Fixed |
| | Variable (PRN or T& E) |
| **Monitoring frequency** (follow up visits) | Every 3 months |
| | Every 2 months |
| | Every month |
| **Cost** (treatment cost compared to current treatment) | 10% Increase |
| | Same cost |
| | 5% Decrease |

PRN, *Pro Re Nata*; T& E, treat-and-extend.

| | Treatment A | Treatment B |
|---|---|---|
| With treatment your **visual function** | ★★★★★ improves by more than 5 letters | ★★★ improves from 1-5 letters |
| With treatment the **presence of intraretinal fluid*** | ═ remains unchanged | ◐↓ is reduced |
| You receive the treatment | in a **variable** regimen according to your results | in a **fixed** regimen |
| You have to go to the retina specialists for your **follow-up visits** | 2018 Every month | 2018 Every month |
| The **total cost of the treatment** compared to your current treatment | ═ Remains the same | ↑ Increases by 10% |
| | I prefer A ○ | I prefer B ○ |

*the presence of fluid in the retina is associated with a greater deterioration of its structure, which may lead to a worse prognosis of the disease.

**Fig 1. Example of hypothetical choice sets presented to participants.**

## Analysis

Patients' sociodemographic and clinical variables were described using absolute and relative frequencies of response for qualitative variables and statistics of centrality and dispersion for the quantitative variables.

The relative importance of each attribute was analyzed using a mixed logit model (Stata software [25]). The mixed logit model assumes that the probability of choosing a profile from a set of alternatives is a function of the attribute levels that characterize the alternatives and a random error term that adjusts for individual-specific variations in preferences [26]. It estimates a coefficient (partial utility) for each attribute level. The statistical significance of a coefficient indicates that the respondents considered the attribute important when making their choices. The sign of a coefficient reflects whether the attribute affects the preference score positively or negatively. The RI of each attribute was calculated as the range of partial utilities for the attribute (difference in partial utilities between the best or most preferred level and the worst or least preferred level of the same attribute), divided by the sum of all ranges across attributes and multiplying by 100.

The MRS was calculated by dividing the partial utility for the attribute levels by the additional costs of the partial utilities.

A subgroup analysis was conducted to compare preferences and MRS between patients and retina specialists. The means of the individual RIs for each group (estimated from the individual partial utilities obtained for each participant) were compared using the Mann-Whitney U test, after verifying that the RI did not present normality. For all the statistical tests, results were considered statistically significant when $p < 0.05$.

## Statement of ethics compliance

This study was conducted according to the principles of the Declaration of Helsinki. It was developed to ensure consistency with the principles of the ICH Harmonized Tripartite Guideline for Good Clinical Practice. The study protocol was submitted to the Spanish Agency of Medicines and Medical Devices. Protocol, informed consent form and other information for patients were approved by the Ethical Committee of Drug Research idcsalud in Catalonia–Hospital General de Catalunya Committee, with ethics approval number 2018/63-OFT-H-UGC. All patients and retina specialists signed a written informed consent form before being included in the study.

## Results

### Characteristics of the study participants

A total of 110 patients [mean age 79.0 years (SD 7.4); 57.3% women] with nAMD [mean years from diagnosis 2.3 years (SD 0.7)] receiving intravitreal anti-VEGF therapy [mean years 2.1 (SD 0.1); 45.5% *Pro Re Nata* (PRN), 44.5% treat-and-extend (T&E), 4.5% fixed and 6.4% other regimen] (S2 Table) and 66 retina specialists working in the SNHS responded to the survey.

All retina specialists (100%) and 95.6% of patients passed the dominant question and therefore were available for analysis.

### Patient and retina specialists' preferences for nAMD treatment characteristics

**Partial utilities.** Partial utilities reflect the importance of an attribute level against a reference level (Ref). Partial utility of the linearly transformed attributes (monitoring frequency and cost) must be interpreted as the importance of 1-unit increases (1 month or 1%, respectively). For patients and retina specialists, partial utilities showed the effect of treatment on

**Table 2. Patient and retina specialists' partial utilities.**

| Attribute | Level | Patients with nAMD | | | Retina specialists | | |
|---|---|---|---|---|---|---|---|
| | | Partial utility | SE | p-value | Partial utility | SE | p-value |
| Effect on visual function | Improvement of more than 5 letters (Ref) | 0.000 | - | - | 0.000 | - | - |
| | Improvement from 1–5 letters | -2.167 | 0.319 | *<0.001* | -2.671 | 0.557 | *<0.001* |
| | Stable | -5.055 | 0.622 | *<0.001* | -7.960 | 1.566 | *<0.001* |
| Effects on retinal fluid | Resolution (Ref) | 0.000 | - | - | 0.000 | - | - |
| | Reduction | -0.569 | 0.255 | *0.026* | -0.869 | 0.449 | 0.053 |
| | No changes | -0.826 | 0.300 | *<0.001* | -1.966 | 0.574 | *0.001* |
| Treatment regimen | Fixed (Ref) | 0.000 | - | - | 0.000 | - | - |
| | Variable (PRN or T& E) | 0.335 | 0.192 | 0.081 | 0.384 | 0.21 | 0.231 |
| Monitoring frequency | Per unit (1 month) | 0.851 | 0.173 | *<0.001* | 2.045 | 0.392 | *<0.001* |
| | Every 3 months | 2.554 | - | - | 6.135 | - | - |
| | Every 2 months | 1.702 | - | - | 4.090 | - | - |
| | Every month | 0.851 | - | - | 2.045 | - | - |
| Cost | Per unit (1%) | -0.033 | 0.021 | 0.119 | -0.047 | 0.036 | 0.184 |
| | Decrease 5% | 0.167 | - | - | 0.236 | - | - |
| | Same cost | 0.000 | - | - | 0.000 | - | - |
| | Increase 10% | -0.333 | - | - | -0.472 | - | - |

(Ref) Reference level.

visual function, on retinal fluid and monitoring requirements as treatment decision-making drivers. Not achieving changes in visual capacity or in the retinal fluid were significantly less preferred (p<0.001), while lower monitoring frequency was preferred (p <0.001). Although not statistically significant (p >0.05), treatments with variable regimens and those with lower cost were preferred (Table 2).

**Relative importance.** The RI of each attribute enables ranking the treatment characteristics and establishing each attribute's importance compared to the rest. Patients and retina specialists gave greater RI to improvements in visual function (Patients: 60.0%; Retina specialists: 52.7%), lower monitoring frequency (Patients: 20.2%; Retina specialists: 27.1%), and reduction in retinal fluid (Patients: 9.8%; Retina specialists: 13.0%), compared to cost and treatment regimen (Fig 2).

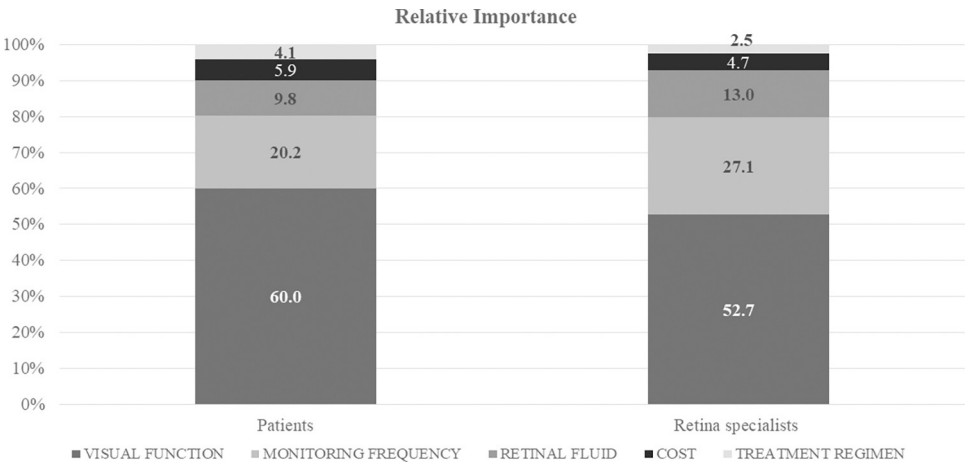

**Fig 2. Relative importance of treatment attributes.**

**Table 3. Differences in marginal rates of substitution (WTP) between patients and retina specialists.**

| Attribute | Level of attribute | WTP (%) | | P value |
|---|---|---|---|---|
| | | Patients | Retina specialists | |
| Visual function | Improve 1–5 letters vs improve > 5 letters | 65.0 | 56.5 | **0,018** |
| Effects on retina fluid | Reduction vs Resolution | 17.1 | 18.4 | 0,7857 |
| Treatment regimen | Fixed vs variable | 10.0 | 8.14 | 0,1961 |
| Monitoring frequency | Increase 1 month | 25.5 | 43.3 | **0,0001** |

**Marginal rate of substitution (willingness to pay).** MRS determines the amount of money that individuals are willing to pay (WTP) to receive their preferred level of an attribute. Patients and retina specialists would agree on an increase in the treatment cost by 65.0% and 56.5%, respectively, in exchange for improvement in visual function and by 25.5% and 43.3% for delaying the frequency of monitoring by one month. For both attributes, differences statistically significant were observed (Table 3).

**Comparison of patient and retina specialists' preferences.** Significant differences were observed in the RI attached to visual function, treatment plan and monitoring frequency. Thus, even though both patients and retina specialists considered improvement in visual function to be the most important attribute, patients gave it greater importance than specialists (p <0.001). A similar trend was observed with regard to the treatment plan, which was more relevant for patients than for retina specialists (p = 0.0122). In contrast, retina specialists attached greater importance to minor monitoring frequency than patients (p<0.001).

## Discussion

Healthcare systems are moving to patient-centered care based on shared decision making [27], whereby both physicians and patients share information, discuss treatment preferences and agree on their treatment plan. Therefore, patient preferences are a growing topic of interest and the number of studies that explore patient priorities and perspectives has increased in the last decade. Preference studies are a means of generating data on patient perceptions and preferences in relation to different aspects of existing or investigational health-related products, services, and interventions [28]. The information gathered in these studies can be used during various stages of clinical care management planning and implementation [29]. Ultimately, patient preference studies can help ensure that healthcare products and services align with patient preferences and needs and that the outcome measures used in assessments are lined up with what is important to patients [30].

Several studies have explored patient preferences for AMD treatment characteristics [16–20, 30, 31]; however, none of them has included other stakeholder perspectives. Since patients and physicians may have different perceptions of the disease and its treatment, the inclusion of retinal specialists in the study allows an assessment of whether clinician preferences are aligned with patient priorities.

In line with other studies, the effect of treatment on visual function and monitoring requirements [16–20, 30, 31] were treatment decision drivers. As expected, given the significant impact vision loss has on patient's independence in activities of daily living [32], achieving good vision was the most important attribute of treatment for patients and retina specialists and, the main treatment decision driver. Patients and retina specialists were willing to accept and increase treatment cost by more than 50% to achieve better visual outcomes. Monitoring frequency was the second decision driver, with lower frequency being preferred. Preferences for treatment monitoring requirements have been evaluated in several studies, with similar

results [16–20, 30, 31]. Data from real world evidence on treatment burden shows that at 2 years approximately 70% of visit intervals (n = 1.344) were ≤ 8 weeks (<4 weeks: 20%; 4–6 weeks: 30%; 6–8 weeks: 14%) while nearly 50% of injection intervals (n = 781) were ≤ 8 weeks (<4 weeks: 4% 4–6 weeks: 26%, 6–8 weeks: 18%) [22]. The burden of treatment related to the need for frequent visits to the hospital may not only have an impact on patients and retinal specialists, but also on caregivers [33]. Monthly monitoring is often difficult to accommodate in retinal units, so reducing monitoring frequency requirements may lower the treatment burden from the retina specialist perspective. Therefore, the greater relative importance and WTP given to this attribute by retina specialists compared to patients is not surprising. Recently, a group of retinal experts in Spain reported that the burden of the disease and monitoring frequency requirements together with organizational and logistic issues are limiting appropriate treatment for nAMD in Spain, especially in patients with flexible regimens [34].

Neovascular AMD is characterized by the presence of choroidal neovascularization resulting in leakage of fluid that accumulates intraretinally or subretinally or below the retinal pigment epithelium [35]. Consequently, clinically, retinal fluid is one of the parameters used to measure disease activity in nAMD, and treatment goals for nAMD include drying the affected eye by inhibiting new blood vessels from leaking fluid and improving or maintaining visual acuity [36]. Our results showed that the third most desirable treatment characteristic was the effect of treatment on retinal fluid. The ranges of preference coefficients illustrated that retina specialists were least sensitive to changes in this attribute, meaning that either reduction or resolution would represent a remarkable achievement from their perspective. Finally, cost and treatment regimens are not treatment drivers. However, in other studies conducted in countries where treatment is not covered by the healthcare system, the cost of treatment is identified as one of the most relevant attributes [16]. This difference may be explained by the influence of treatment cost on preferences may be reduced when the cost is covered by an insurance provider or healthcare system, as is in Spain.

The study has some limitations, most of them inherent to conjoint analysis methodology. Although conjoint analysis represents one of the most robust and widely used approaches to assessing patient preferences for treatment characteristics, there is always the risk of a gap between stated and revealed preferences [21]. Since the number of attributes or levels presented in a DCE is limited, we selected the most relevant attributes using focus groups with patients and retina specialists. Still, this careful procedure does not guarantee that attributes not included may also be relevant and play a role in treatment decision-making. A particular strength of this methodology is that the expression of a preference in the form of a choice can be performed even by those with mild to moderate cognitive impairment. It is important to keep in mind that the study was conducted in Spain, and its results should be interpreted in their context and may not apply to other countries. The last limitation is related to the selection of participants. At the time of the study, patients had been on anti-VEGF treatment for 24 months, so we cannot exclude the possibility that their prior experiences with anti-VEGF treatment may have biased patients' preferences. Hence, the preferences of intravitreal treatment naïve patients might differ from the ones we observed. Moreover, data regarding sociodemographic characteristics of retina specialists were not collected during the study.

## Conclusion

The study results provide relevant information regarding patient and retina specialists' preferences for nAMD treatment that may contribute to guiding treatment decisions. For patients and retina specialists, the election of a treatment option is determined by the ability of the treatment to improve visual function. The treatment monitoring requirements are also

considered during the election of a treatment, mainly from a retina specialist perspective. The use of more efficacious anti-VEGF agents (gains in visual function and better anatomical outcomes) with a longer duration of action (minor monitoring requirements) may align treatment characteristics with patients' and specialists' preferences. A better alignment with patients and retina specialists' preferences would facilitate better disease management, reducing the burden on patients and their caregivers and the use of healthcare resources, thus fulfilling the unmet needs of patients and retina specialists.

## Supporting information

**S1 Table. Patients' sociodemographic and clinical characteristics.**
(DOCX)

**S2 Table. List of hospital / institutions participating in the study.**
(DOCX)

**S1 Dataset.**
(ZIP)

## Acknowledgments

These results have been presented at the Virtual International Society for Pharmacoeconomics and Outcomes Research (ISPOR) Europe (30 november– 3 december 2020).

The authors would like to thank to all experts who participated in the focus group (Pablo L from the Department of Ophthalmology of the Hospital Universitario Miguel Servet; Escobar JJ from Hospital Dos de Maig of Barcelona; Sararols L from the Hospital General de Catalunya of Barcelona; López-Garrido JA from the Department of Ophthalmology of Hospital Galda-kao-Usansolo of Bilbao; Farfan FJ from the European University of Madrid; Gámez MJ from the Hospital de la Santa Creu i Sant Pau of Barcelona; and patients with AMD from the patient advocacy group (Mácula-Retina) and all the centres investigators that have collaborated in the AMD-MANAGE study: Dr. Maximino José Abraldes López-Veiga (Complexo Hospitalario Universitario de Santiago de Compostela), Dr. Rodrigo Abreu González (Hospital Universitario Nuestra Señora de Candelaria), Dr. Daniel Aliseda Pérez de Madrid (Complejo Hospitalario de Navarra), Dr. Enrique Cervera Taulet (Hospital General Universitario de Valencia), Dr. José Ignacio Fernández-Vigo Escribano (Hospital Clínico San Carlos), Dr. Gonzaga Garay Aramburu (Hospital Universitario Araba), Dr. Saturnino Manuel Gismero Moreno (Hospital Costa del Sol), Dr. María Jesús Huertos Carrillo (Hospital Universitario Puerto Real), Dr. Francisco Javier Lavid de los Mozos (Hospital Punta de Europa), Dr. María Isabel López Gál-vez (Hospital Clínico Universitario de Valladolid), Dr. José Antonio López Garrido (Hospital Universitario de Galdakao), Dr. Maria del Carmen López Quero (Hospital Virgen de la Arrix-aca), Dr. José Luis Olea Vallejo (Hospital Universitari Son Espases), Dr. Pere Romero Aroca (Hospital Universitari Sant Joan de Reus), Dr. Oscar Ruiz Moreno (Hospital Universitario Miguel Servet), Dr. Laura Sararols Ramsay (Hospital General de Granollers), Dr. Alicia Trave-set Maeso (Hospital Universitari Arnau de Vilanova).

## Author Contributions

**Conceptualization:** Roberto Gallego-Pinazo, Begoña Pina-Marin, Marta Comellas, Susana Aceituno, Laia Gómez-Baldó, Carles Blanch.

**Data curation:** Roberto Gallego-Pinazo, Begoña Pina-Marin.

**Formal analysis:** Susana Aceituno.

**Methodology:** Marta Comellas.

**Writing – original draft:** Marta Comellas.

**Writing – review & editing:** Roberto Gallego-Pinazo, Begoña Pina-Marin, Susana Aceituno, Laia Gómez-Baldó, Carles Blanch.

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
