## [Decision Letter · Decision Letter 0]

14 Sep 2021

PONE-D-21-13952

Patient and retina specialists’ preferences in neovascular age-related macular degeneration

PLOS ONE

Dear Dr. Blanch,

Thank you for submitting your manuscript to PLOS ONE. After careful consideration, we feel that it has merit but does not fully meet PLOS ONE’s publication criteria as it currently stands. Therefore, we invite you to submit a revised version of the manuscript that addresses the points raised during the review process.

This study considers preferences for treatment for neovascular age-related macular degeneration (nAMD) from both a patient and retinal practitioner perspective, using .This involved a series of hypothetical options for treatment, followed by focus groups. This does provide good background for expectations in ongoing personalised nAMD treatment but there are several areas that require further details and clarification.

1. For the selection of retinal specialists, please provide a brief description of the AMD-MANAGE study and criteria for this study, and how specialists for the current study were selected.

2. Informed consent is noted but was a Human Ethics Committee aproval involved? Please provide further details, thank you. Please also provide full details on patient and specialist demographics (e.g. patterns of treatment and monitoring, how long receiving treatment etc). Thank you,

3. In the Abstract and Conclusions, the authors indicate that "these results suggest that the use of more efficacious anti-VEGF agents with a longer duration of action.." are required. Please note that this aspect was not assessed in the study and should be amended.

4. Please provide more details on the focus group aspects, including overall M:F for the group. All patients were females and age range 45 to 75 - were these new patients to anti-VEGF therapies, what were their experiences? Same for the retina specialists and pharmacists. Thank you.

5. Please address all the comments provided by the reviewers.

We look forward to receiving your revised manuscript.

Kind regards,

Michele Madigan

Academic Editor

PLOS ONE

Journal Requirements:

2. Please provide additional details regarding the retina specialists participant consent. In the ethics statement in the Methods and online submission information, please ensure that you have specified (1) whether consent was suitably informed and (2) what type you obtained (for instance, written or verbal). If your study included minors under age 18, state whether you obtained consent from parents or guardians. If the need for consent was waived by the ethics committee, please include this information.

3. Please include in your Methods section (or in Supplementary Information files) the participating hospitals/institutions.

4. Thank you for stating the following in the Competing Interests/Financial Disclosure* (delete as necessary) section:

“I have read the journal's policy and the authors of this manuscript have the following competing interests: LGB and CB are paid employees of Novartis.”

We note that one or more of the authors are employed by a commercial company:  Novartis

6. We noticed you have some minor occurrence of overlapping text with the following previous publication(s), which needs to be addressed:

- https://linkinghub.elsevier.com/retrieve/pii/S0161642020300415

- https://www.dovepress.com/the-emotional-and-physical-impact-of-wet-age-related-macular-degenerat-peer-reviewed-fulltext-article-OPTH

- https://linkinghub.elsevier.com/retrieve/pii/S1098301520342388

In your revision ensure you cite all your sources (including your own works), and quote or rephrase any duplicated text outside the methods section. Further consideration is dependent on these concerns being addressed.

Reviewers' comments:

Reviewer's Responses to Questions

**Comments to the Author**

1. Is the manuscript technically sound, and do the data support the conclusions?

Reviewer #1: No

Reviewer #2: Yes

2. Has the statistical analysis been performed appropriately and rigorously? 

Reviewer #1: I Don't Know

Reviewer #2: I Don't Know

3. Have the authors made all data underlying the findings in their manuscript fully available?

Reviewer #1: Yes

Reviewer #2: No

4. Is the manuscript presented in an intelligible fashion and written in standard English?

Reviewer #1: No

Reviewer #2: Yes

5. Review Comments to the Author

Reviewer #1: Patient and retinal specialist preferences for nAMD treatment is an interesting and relevant study. The authors should state earlier in the manuscript how this study stands apart from other patient preference studies. Also becareful of concluding that better visual & anatomical outcomes would be achieved with longer acting anti-VEGF agents. This is not what you analysed.

Abstract:

line 29 define VEGF

line34 Methods: include who the participants are ?nAMD patients on anti-VEGF treatment, previous treatment?

Abstract Results

line 40 need to say aged 79 or 79 years old

line 42 insert 'their' in front of 'visual function'

line 45 improvements of visual function (instead of improving)

line 45 agree on instead of 'for delaying'

line 46 when is the frequency delayed? with or without treatment

line 47 Conclusion : first sentence does not make sense

line 50 FInal line of conclusion is wrong: These results do not suggest that more efficacious anti-VEGF agents with a longer duration of action would facilitate better disease management. This study is about patient and doctor preference.

Introduction

line 2 age-related not 'aged-related'

line 6 'by' instead in 2040

line 8 ameliorate isn't quite the correct word to describe this means 'make better'

line 14 In this study, prior to 'the most commonly reported diseae'

line 18 insert 'their' prior to vision impairment

line 22 treatment should be inserted after anti-VEGF

line 22 visual decline not 'visual damage'

line 24 last line does not make sense

line 31,32 and 33 is not clear

line 22 new paragraph in front of Conjoint analysis

You need to explain in the introduction why this analysis is different/better than others nAMD studies completed in this area

line 43 who are the stakeholders'? Pharma companies?

METHODOLOGY

Methods need to define what is ment by treatment regimen and monitoring frequency and how they differ. For instance if fix regimen then is monitoring the same time as treatment? more detail required here

Line 54 aret the retinal specialists independent or do they treat the patients in the study?

I would mention ethics approval in first paragraph of methods

line 72 define the acronyms

Table 2 needs to be translated into English

line 116 RI is used earlier in the manuscript and should be abbreviated defined when first used

line 120 MRS previously defined

Results

line 137 79.0 uears

line 138 2.3 years

line 140 has SNHS been defined

line 150 sentence not clear

line 160 ? in contrast to cost and treatment regimen

line 165 is the cost to the patient or to the retinal specialist or both not defined

line 177 monitoring frequency, is this increasing or decreasing?

Line 180 who is 'we'? Your centre, your country? international retinal specialist?

line 181 their instead of 'a' treatment plan

line 198 insert 'patient's' before independence

line 200 remove 'therefore, the main treatment decision driver.'

Other bias is patients that started treatment but couldn't continue due to cost or other. Not included in the group.

The conclusions again are to clear. How can a study on patient and treating specialist's preference concluded that longer acting anti-VEGF agents would facilitate better disease management (visual function and anatomical outcomes). That may be true but not what the authors have studied here.

Reviewer #2: A unique, well-written study that contrasts the differing priorities of patients and retinal specialists with respects to anti-VEGF treatment in neovascular AMD. Certainly, a key strength of the work is the juxtaposition of findings from different stakeholder groups.

Lines 19-21: Treatment accessibility (distance to treatment, time spent in the clinic, caregiver demands) and patient understanding of the benefits of treatment are also important barriers that should be considered, either here or elsewhere.

Lines 48-51: Can the authors justify their 2 year inclusion criteria? Acknowledged in part on lines 238-242, patients on treatment for 2 years may have quite different preferences compared to patients newer to treatment and arguably, it is the second population group with lower adherence, which requires more study. On this point, it is also worth commenting on the participant characteristics in more detail generally.

All patients were recruited from tertiary care so presumably proceeding with anti-VEGF treatment, which again, would lead them to have significantly different preferences to treatment naïve patients or patients considering treatment discontinuation. How might retinal specialists working in the Spanish National Healthcare System compare to retinal specialists in other countries?

Line 71: What search terms were used?

Line 76-77: Further detail on the composition and conduct of these focus groups is important to include. Was it a single focus group with 10 participants total? How were the participants identified? How was data collected? Was there an interview guide? And how was the data analysed? Did all participants agree on the final list of attributes? And if so, how was consensus reached? Interesting that access (distance to the treating clinic) was not considered an important attribute to include.

Line 82: How were the attribute levels determined? E.g., the levels for the attribute effect on visual function seem very limited. Presumably, they refer to high contrast visual acuity but as we know, late AMD also causes central scotomas, the size and severity of these scotomas might have also been considered. For patients, the impact on activities of daily living would rate quite differently, especially depending on how the activity is framed. This seems essential information to include as it forms the basis of the options presented. Other considerations include how fluid reduction/resolution was communicated. Does the latter mean zero fluid tolerance? Cost level options were defined relative to current cost. This is discussed in part in the manuscript discussion but any attempts to minimize bias should be presented in the methods.

Line 99: How long did the survey take to complete and how was it administered? In hard copy? What is the likelihood of satisficing, response bias or the Hawthorne effect?

Is an English version of Table 2 available? The wording used to ask participants about their preferences is very important.

Line 209: Do you have any data on how the participants attended their treatment or monitoring appointments? Patients more reliant on their carers might have expressed different preferences relative to more independent patients.

6. PLOS authors have the option to publish the peer review history of their article (what does this mean?). If published, this will include your full peer review and any attached files.

Reviewer #1: No

Reviewer #2: No

---

## [Author Response · Author response to Decision Letter 0]

18 Oct 2021

Reviewer #1: Patient and retinal specialist preferences for nAMD treatment is an interesting and relevant study. The authors should state earlier in the manuscript how this study stands apart from other patient preference studies. 

Thank you for your comments. Several studies have explored patients' preferences regarding nAMD treatment; however, none have included perspectives of retina specialists. The inclusion of both views, patients and retina specialists, is crucial to understand disease management. 

This information has been included in the introduction.

Line 46. "Although several studies have explored preferences from patients with AMD perspective, none of them have included the retina specialist perspective. Given the high burden of managing AMD for both patients and retina specialists, explore different stakeholders' preferences is crucial to understanding disease management. Therefore, this study aims to elicit preferences for treatment characteristics in nAMD by including the perspectives of both patients and retina specialists.

Also becareful of concluding that better visual & anatomical outcomes would be achieved with longer acting anti-VEGF agents. This is not what you analysed.

According to editor and reviewer comment, abstract and conclusions have been modified:

Line 51. " Efficacy of treatment, in terms of visual function improvements, is the main driver for treatment election for both patients and retina specialists. Treatment monitoring requirements are also considered, mainly from the retina specialist's perspective. These results suggest that the use of more efficacious anti-VEGF agents with a longer duration of action may contribute to aligning treatment characteristics with patients/specialists’ preferences. A better alignment would facilitate better disease management, fulfilling the unmet needs of patients and retina specialists.”

Line 282. "The use of more efficacious anti-VEGF agents (gains in visual function and better anatomical outcomes) with longer duration of action (minor monitoring requirements) may align treatment characteristics with patients' and specialists' preferences. A better alignment with patients and retina specialists' preferences would facilitate better disease management, reducing the burden on patients and their caregivers and the use of healthcare resources, thus fulfilling the unmet needs of patients and retina specialists. "

Abstract:

line 29 define VEGF

Done 

line34 Methods: include who the participants are ?nAMD patients on anti-VEGF treatment, previous treatment?

Patients with nAMD receiving anti-VEGF drugs for at least 2 years without previous experience with anti-VEGF were included in the study. 

This information is included in the abstract and methodology. 

Line 35. "Method: A discrete choice experiment was conducted. Participants (patients > 50 years with nAMD receiving anti-VEGF drugs for at least 2 years and without previous experience with anti-VEGF and retina specialists working in the Spanish National Healthcare System) were asked to select one of two hypothetical treatments resulting from the combination of five attributes (effects on visual function, effects on retinal fluid, treatment regimen, monitoring frequency, and cost); their levels were identified by reviewing the literature and two focus groups."

Line 57. " Patients over the age of 50 years with nAMD receiving anti-VEGF drugs for at least 2 years and retina specialists working in the Spanish National Healthcare System (SNHS) were invited to participate in the study. They were selected from the universe of the AMD-MANAGE study (patients recruited from 20 public and private tertiary hospitals from different Spanish regions that met the following selection criteria: adult naïve patients ⩾50 years with confirmed nAMD diagnosis who started anti-VEGF treatment between November 1st, 2016 and February 28th, 2017, with a follow-up of 24 months and not participating in any other clinical study) (S1 Appendix). [23] ."

Abstract Results

line 40 need to say aged 79 or 79 years old

Done

line 42 insert 'their' in front of 'visual function'

Done

line 45 improvements of visual function (instead of improving)

Done

line 45 agree on instead of 'for delaying'

Done

line 46 when is the frequency delayed? with or without treatment

All the patients included were treated (by inclusion criteria) and the frequency of delay was set on at least one month. 

line 47 Conclusion : first sentence does not make sense

According to reviewer suggestion sentence has been rewriting: 

Line 51: “Efficacy of treatment, in terms of visual function improvements, is the main driver for treatment election for both patients and retina specialists.”

line 50 FInal line of conclusion is wrong: These results do not suggest that more efficacious anti-VEGF agents with a longer duration of action would facilitate better disease management. This study is about patient and doctor preference.

These sentences have been modified:

Line 64. “These results suggest that the use of more efficacious anti-VEGF agents with a longer duration of action may contribute to aligning treatment characteristics with patients/specialists’ preferences. A better alignment would facilitate better disease management, fulfilling the unmet needs of patients and retina specialists.”

Introduction

line 2 age-related not 'aged-related'

Done

line 6 'by' instead in 2040

Done

line 8 ameliorate isn't quite the correct word to describe this means 'make better'

Ameliorate has been changed for reduce

line 14 In this study, prior to 'the most commonly reported diseae'

Done

line 18 insert 'their' prior to vision impairment

Done

line 22 treatment should be inserted after anti-VEGF

Done

line 22 visual decline not 'visual damage'

Done

line 24 last line does not make sense

Following the reviewer comment, the sentence has been rewritten: 

Line 24. “However, the need for frequent monitoring visits and intravitreal injections lead a significant burden on patients, caregivers and retina specialists.”

line 31,32 and 33 is not clear

The sentence has been modified:

Line 32. "The promotion of shared decision-making and incorporation of patient preferences in the disease management decision could improve the effectiveness of healthcare interventions by increasing patient satisfaction and improving adherence to treatments"

line 22 new paragraph in front of Conjoint analysis

You need to explain in the introduction why this analysis is different/better than others nAMD studies completed in this area

The following information has been included.

Line 46. "Although several studies have explored preferences from patients with AMD perspective, none of them have included the retina specialist perspective. Given the high burden of managing AMD for both, patients and retina specialist, explore different stakeholders' preferences is crucial to understanding disease management."

line 43 who are the stakeholders'? Pharma companies?

No, stakeholders refer to patients, and different healthcare professionals involved in disease management. 

METHODOLOGY

Methods need to define what is ment by treatment regimen and monitoring frequency and how they differ. For instance if fix regimen then is monitoring the same time as treatment? more detail required here

Definition of attributes is included in Table 1. 

Table 1. Attributes and levels included in the DCE

Attribute Level

Effect on visual function (best-corrected visual acuity improvements from baseline) Stable (no changes)

Improvement of more than 5 letters

Improvement from 1-5 letters

Effects on retinal fluid (change in intraretinal fluid volume) Reduction

Resolution

No changes

Treatment regimen (treatment plan included dosage, schedule, and duration of treatment) Fixed

Variable (PRN or T& E)

Monitoring frequency (follow up visits) Every 3 months

Every 2 months

Every month

Cost (treatment cost compared to current treatment) 10% Increase 

Same cost

5% Decrease 

Line 54 aret the retinal specialists independent or do they treat the patients in the study?

Most of retinal specialists treat patients in the study. However, some of them do not. 

I would mention ethics approval in first paragraph of methods

Following journal guidelines, ethics approval is mentioned in the Statement of Ethics Compliance. 

line 72 define the acronyms

Definitions of acronyms have been included.

Line 81. " A literature review was conducted to identify the potential attributes and levels to be included in the DCE. Key terms related to the disease (“age-related macular degeneration”, “ADM”, macular degeneration [MeSH]), treatment (“treatment”, “medication) and stated-preferences studies (“conjoint”, “conjoint analysis”, “conjoint measurement”, “conjoint studies”, “discrete choice experiments”, “DCE”, “discrete choice modeling”, “preference studies”, Patient Preferences [MeSH]) joined by Booleans operators “or” and “and” were used to search in MedLine/PubMed, Cochrane Library, Institute for Scientific Information Web Of Knowledge (ISI WOK) and SCOPUS databases.”

Table 2 needs to be translated into English

Following reviewer comment, Table 2 has been translated into English. 

line 116 RI is used earlier in the manuscript and should be abbreviated defined when first used

According to the reviewer comment, relative importance has been defined when first used

Line 77. The DCE results provide information on the relative importance (RI) of the different attributes and the rate at which respondents are willing to trade one attribute for preferred levels of another attribute (marginal rates of substitution, MRS) [25].

line 120 MRS previously defined

Since MRS has been defined previously, the definition of MRS has been deleted. 

Results

line 137 79.0 uears

Done

line 138 2.3 years

Done

line 140 has SNHS been defined

SNHS has been defined previously (line 56)

line 150 sentence not clear

According to the reviewer comment, the sentence has been rewritten:

Line 178. “Not achieving changes in visual capacity or in the retinal fluid were significantly less preferred (p<0.001), while lower monitoring frequency was preferred (p <0.001).”

line 160 ? in contrast to cost and treatment regimen

The sentence has been modified according to reviewer comments:

Line 190. “Patients and retina specialists gave greater RI to improvements in visual function (Patients: 60.0%; Retina specialists: 52.7%), lower monitoring frequency (Patients: 20.2%; Retina specialists: 27.1%), and reduction in retinal fluid (Patients: 9.8%; Retina specialists: 13.0%), compared to cost and treatment regimen (Figure 1).”

line 165 is the cost to the patient or to the retinal specialist or both not defined

It refers to treatment cost. The sentence has been modified accordingly. 

Line 194. "Patients and retina specialists would agree on an increase in the treatment cost by 65.0% and 56.5%, respectively, in exchange for improvement in visual function and by 25.5% and 43.3% for delaying the frequency of monitoring by one month."

line 177 monitoring frequency, is this increasing or decreasing?

It refers to less monitoring frequency. The sentence has been modified

Line 206. "In contrast, retina specialists attached greater importance to minor monitoring frequency than patients (p<0.001)."

Line 180 who is 'we'? Your centre, your country? international retinal specialist?

It refers to the Healthcare systems.

The sentence has been modified accordantly.

Line 201. “Healthcare systems are moving to patient-centered care based on shared decision making [28], whereby both physicians and patients share information, discuss treatment preferences and agree on their treatment plan.”

line 181 their instead of 'a' treatment plan

Done

line 198 insert 'patient's' before independence

Done

line 200 remove 'therefore, the main treatment decision driver.'

Done

Other bias is patients that started treatment but couldn't continue due to cost or other. Not included in the group.

It is important to notice that in Spain, these treatments are reimbursed by the nation health service (NHS); therefore, patients do not need to pay for them. 

The conclusions again are to clear. How can a study on patient and treating specialist's preference concluded that longer acting anti-VEGF agents would facilitate better disease management (visual function and anatomical outcomes). That may be true but not what the authors have studied here.

Conclusions of the study has been rewritten in order to clarify this aspect

Line. 283" The study results provide relevant information regarding patient and retina specialists’ preferences for nAMD treatment that may contribute to guiding treatment decisions. For patients and retina specialists, the election of a treatment option is determined by the ability of the treatment to improve visual function. The treatment monitoring requirements are also considered during the election of a treatment, mainly from a retina specialist perspective. The use of more efficacious anti-VEGF agents (gains in visual function and better anatomical outcomes) with a longer duration of action (minor monitoring requirements) may align treatment characteristics with patients’ and specialists’ preferences. A better alignment with patients and retina specialists’ preferences would facilitate better disease management , reducing the burden on patients and their caregivers and the use of healthcare resources, thus fulfilling the unmet needs of patients and retina specialists."

Reviewer #2: A unique, well-written study that contrasts the differing priorities of patients and retinal specialists with respects to anti-VEGF treatment in neovascular AMD. Certainly, a key strength of the work is the juxtaposition of findings from different stakeholder groups.

Lines 19-21: Treatment accessibility (distance to treatment, time spent in the clinic, caregiver demands) and patient understanding of the benefits of treatment are also important barriers that should be considered, either here or elsewhere.

We agree with reviewer comments, however, in order to not extend the introduction, only those aspects were included that, according to the authors, were most relevant from the patient’s and caregiver’s perspective. 

Lines 48-51: Can the authors justify their 2 year inclusion criteria? Acknowledged in part on lines 238-242, patients on treatment for 2 years may have quite different preferences compared to patients newer to treatment and arguably, it is the second population group with lower adherence, which requires more study. On this point, it is also worth commenting on the participant characteristics in more detail generally.

Patients with at least two years of experience with anti-VEGF were selected to ensure that they understand the treatment burden, and therefore, can make an informed decision regarding treatment election. 

All patients were recruited from tertiary care so presumably proceeding with anti-VEGF treatment, which again, would lead them to have significantly different preferences to treatment naïve patients or patients considering treatment discontinuation. 

We agree with the reviewer, and the point had been addressed in the study limitation.

Line 274. "Hence, the preferences of intravitreal treatment naïve patients might differ from the ones we observed."

How might retinal specialists working in the Spanish National Healthcare System compare to retinal specialists in other countries?

We agree with the reviewer that since the study was conducted in Spain, the results should be interpreted in their context and may not apply to other settings. Therefore, this limitation has been included in the study limitation.

Line 269. "It is important to keep in mind that the study was conducted in Spain, and its results should be interpreted in their context and may not apply to other countries."

Line 71: What search terms were used?

According to the reviewer comment, more details regarding literature review were included. 

Line 81. "A literature review was conducted to identify the potential attributes and levels to be included in the DCE. Key terms related to the disease ("age-related macular degeneration", "ADM", macular degeneration [MeSH]), treatment ("treatment", "medication) and stated-preferences studies ("conjoint", "conjoint analysis", "conjoint measurement", "conjoint studies", "discrete choice experiments", "DCE", "discrete choice modeling", "preference studies", Patient Preferences [MeSH]) joined by Booleans operators "or" and "and" were used to search in MedLine/PubMed, Cochrane Library, Institute for Scientific Information Web Of Knowledge (ISI WOK) and SCOPUS databases. Studies published until May 2018 that assessed patient or retina specialists' preferences for AMD treatment attributes and/or their willingness to pay for gaining health benefits or avoiding side effects were selected."

Line 76-77: Further detail on the composition and conduct of these focus groups is important to include. Was it a single focus group with 10 participants total? How were the participants identified? How was data collected? Was there an interview guide? And how was the data analysed? Did all participants agree on the final list of attributes? And if so, how was consensus reached? Interesting that access (distance to the treating clinic) was not considered an important attribute to include.

According to the reviewer suggestion, more details regarding focus groups were included:

Line 91. " Two focus groups to define the set of attributes and levels to be included in the DCE were conducted, one with patients (n=4 patients, 100% women, range age 45 to 70 years, 100% in anti-VEG treatment; S2 Table) and one with experts in AMD management (n=4 retina specialists and n=2 hospital pharmacists; S2 Table). Patients with AMD who participated in the focus group were identified by the patient advocacy group (Mácula Retina); the study coordinator selected experts based on their expertise in ADM management. Participants in the focus groups discussed the validity and relevance of the potential attributes and levels identified in the literature review. Moreover, they completed the list with those attributes and levels not previously described in the literature but important from their perspective. Attributes were ranked from most to least important based on their preferences. The interpretation of the qualitative analysis and the analysis of the ranking exercises allowed to narrow down the list of attributes. Additionally, attributes and levels were tested to check for any problems in interpretation and face validity. 

During the focus group with experts, consensus regarding the attributes/levels to be included in the DCE was achieved. As a result, a total of 5 attributes were selected based on their relevance from patients’ and experts’ perspectives. Following ISPOR recommendation, levels of each attribute were limited to four, extreme values that may cause a grounding effect or the use of ranges to define attributes were avoided, and unplausible profiles were excluded [22] (Table 1).”

Line 82: How were the attribute levels determined? E.g., the levels for the attribute effect on visual function seem very limited. Presumably, they refer to high contrast visual acuity but as we know, late AMD also causes central scotomas, the size and severity of these scotomas might have also been considered. For patients, the impact on activities of daily living would rate quite differently, especially depending on how the activity is framed. This seems essential information to include as it forms the basis of the options presented. Other considerations include how fluid reduction/resolution was communicated. Does the latter mean zero fluid tolerance? Cost level options were defined relative to current cost. This is discussed in part in the manuscript discussion but any attempts to minimize bias should be presented in the methods.

Following reviewer comment, more information related to level selection was included:

Line 91. " Two focus groups to define the set of attributes and levels to be included in the DCE were conducted, one with patients (n=4 patients, 100% women, range age 45 to 70 years, 100% in anti-VEG treatment; S2 Table) and one with experts in AMD management (n=4 retina specialists and n=2 hospital pharmacists; S2 Table). Patients with AMD who participated in the focus group were identified by the patient advocacy group (Mácula Retina); the study coordinator selected experts based on their expertise in ADM management. Participants in the focus groups discussed the validity and relevance of the potential attributes and levels identified in the literature review. Moreover, they completed the list with those attributes and levels not previously described in the literature but important from their perspective. Attributes were ranked from most to least important based on their preferences. The interpretation of the qualitative analysis and the analysis of the ranking exercises allowed to narrow down the list of attributes. Additionally, attributes and levels were tested to check for any problems in interpretation and face validity. 

During the focus group with experts, consensus regarding the attributes/levels to be included in the DCE was achieved. As a result, a total of 5 attributes were selected based on their relevance from patients’ and experts’ perspectives. Following ISPOR recommendation, levels of each attribute were limited to four, extreme values that may cause a grounding effect or the use of ranges to define attributes were avoided, and unplausible profiles were excluded [22] (Table 1).”

Additionally, it is important to notice that one of the study limitations described:

Line 264. "Since the number of attributes or levels presented in a DCE is limited; we selected the most relevant attributes using focus groups with patients and retina specialists. Still, this careful procedure does not guarantee that attributes not included may also be relevant and play a role in treatment decision-making."

Line 99: How long did the survey take to complete and how was it administered? In hard copy? What is the likelihood of satisficing, response bias or the Hawthorne effect?

A paper questionnaire was administered. 

Is an English version of Table 2 available? The wording used to ask participants about their preferences is very important.

Table 2 has been translated to English. 

Line 209: Do you have any data on how the participants attended their treatment or monitoring appointments? Patients more reliant on their carers might have expressed different preferences relative to more independent patients.

Unfortunately, this information was not collected during the study.

---

## [Decision Letter · Decision Letter 1]

29 Nov 2021

PONE-D-21-13952R1Patient and retina specialists’ preferences in neovascular age-related macular degenerationPLOS ONE

Dear Dr. Blanch,

Thank you for submitting your manuscript to PLOS ONE. After careful consideration, we feel that it has merit but does not fully meet PLOS ONE’s publication criteria as it currently stands. Therefore, we invite you to submit a revised version of the manuscript that addresses the points raised during the review process .Thank you for the revised manuscript and fr addressing the reviewers comments. A few minor comments below;

p.58 (track changes copy): please explain what a 'naive' adult patient refers to in the context of the current study. The term is used in several places throughout the manuscript.

The reviewer also noted minor typographical errors that should be edited

The Supporting Information data provided could also presented with Column titles in English if possible. Thank you.

We look forward to receiving your revised manuscript.

Kind regards,

Michele Madigan

Academic Editor

PLOS ONE

Journal Requirements:

Reviewers' comments:

Reviewer's Responses to Questions

**Comments to the Author**

1. If the authors have adequately addressed your comments raised in a previous round of review and you feel that this manuscript is now acceptable for publication, you may indicate that here to bypass the “Comments to the Author” section, enter your conflict of interest statement in the “Confidential to Editor” section, and submit your "Accept" recommendation.

Reviewer #2: All comments have been addressed

2. Is the manuscript technically sound, and do the data support the conclusions?

Reviewer #2: Yes

3. Has the statistical analysis been performed appropriately and rigorously? 

Reviewer #2: I Don't Know

4. Have the authors made all data underlying the findings in their manuscript fully available?

Reviewer #2: Yes

5. Is the manuscript presented in an intelligible fashion and written in standard English?

Reviewer #2: Yes

6. Review Comments to the Author

Reviewer #2: All comments have been addressed and the manuscript improved.

Some minor outstanding notes:

Line 58 (track changes copy): What is an "adult naive" patient? Have the authors included the word naive here by mistake?

There is a repeating typographical error (ADM should be AMD) e.g. on lines 82, 97 and elsewhere

The authors have included their data in the supporting information but it is not in English

7. PLOS authors have the option to publish the peer review history of their article (what does this mean?). If published, this will include your full peer review and any attached files.

Reviewer #2: No

---

## [Author Response · Author response to Decision Letter 1]

9 Dec 2021

Dear Editor and Reviewers:

We enclose a revised version of the manuscript "Patient and retina specialists' preferences in neovascular age-related macular degeneration treatment. A Discrete Choice Experiment.". 

We appreciate the helpful observations and valuable suggestions of the reviewers. We have addressed the reviewer' comments in the revised manuscript. Modifications are highlighted in track changes. Our responses are listed below (in red) following each specific comment.

 

Reviewers' comments:

Some minor outstanding notes:

Line 58 (track changes copy): What is an "adult naive" patient? Have the authors included the word naive here by mistake?

In this context, an adult naïve is a patient with nAMD whit no previous therapeutic exposure to anti-VEGF treatment. 

We have defined naïve patients in the manuscript:

Line 52: “adult naïve (no previous exposure to anti-VEGF treatment) patients ⩾50 years with confirmed nAMD diagnosis who started anti-VEGF treatment between November 1st, 2016 and February 28th, 2017, with a follow-up of 24 months and not participating in any other clinical study)”

There is a repeating typographical error (ADM should be AMD) e.g. on lines 82, 97 and elsewhere

Typographical errors have been corrected. 

The authors have included their data in the supporting information but it is not in English

Supporting information has been translated into English.

---

## [Editor Report · Decision Letter 2]

15 Dec 2021

Patient and retina specialists' preferences in neovascular age-related macular degeneration treatment. A discrete Choice Experiment.

PONE-D-21-13952R2

Dear Dr. Blanch,

We’re pleased to inform you that your manuscript has been judged scientifically suitable for publication and will be formally accepted for publication once it meets all outstanding technical requirements.

Kind regards,

Michele Madigan

Academic Editor

PLOS ONE

Additional Editor Comments (optional):

Thank you for addressing the reviewer comments.
---

## [Editor Report · Acceptance letter]

22 Dec 2021

PONE-D-21-13952R2 

Patient and retina specialists’ preferences in neovascular age-related macular degeneration treatment. A Discrete Choice Experiment. 

Dear Dr. Blanch:

I'm pleased to inform you that your manuscript has been deemed suitable for publication in PLOS ONE. Congratulations! Your manuscript is now with our production department. 

Kind regards, 

on behalf of

Dr. Michele Madigan 

Academic Editor

PLOS ONE